# Characterization of Excited-State Electronic Structure in Diblock π-Conjugated Oligomers with Adjustable Linker Electronic Coupling

**DOI:** 10.3390/molecules29112678

**Published:** 2024-06-05

**Authors:** Habtom B. Gobeze, Muhammed Younus, Michael D. Turlington, Sohel Ahmed, Kirk S. Schanze

**Affiliations:** Department of Chemistry, University of Texas at San Antonio, One UTSA Circle, San Antonio, TX 78249, USA; habtom.gobeze@utsa.edu (H.B.G.); myounus.che@gmail.com (M.Y.); michaelturlington@gmail.com (M.D.T.); sohel.ahmed@utsa.edu (S.A.)

**Keywords:** diblock oligomer, photophysics, charge transfer, energy transfer, oligothiophene, benzothiadiazole, transient absorption

## Abstract

Diblock conjugated oligomers are π-conjugated molecules that contain two segments having distinct frontier orbital energies and HOMO-LUMO gap offsets. These oligomers are of fundamental interest to understand how the distinct π-conjugated segments interact and modify their excited state properties. The current paper reports a study of two series of diblock oligomers that contain oligothiophene (T_n_) and 4,7-bis(2-thienyl)-2,1,3-benzothiadiazole (TBT) segments that are coupled by either ethynyl (-C≡C-) or *trans*-(-C≡C-)_2_Pt(II)(PBu_3_)_2_ acetylide linkers. In these structures, the T_n_ segment is electron rich (donor), and the TBT is electron poor (acceptor). The diblock oligomers are characterized by steady-state and time-resolved spectroscopy, including UV-visible absorption, fluorescence, fluorescence lifetimes, and ultrafast transient absorption spectroscopy. Studies are compared in several solvents of different polarity and with different excitation wavelengths. The results reveal that the (-C≡C-) linked oligomers feature a delocalized excited state that takes on a charge transfer (CT) character in more polar media. In the (-C≡C-)_2_Pt(II)(PBu_3_)_2_-linked oligomers, there is weak coupling between the T_n_ and TBT segments. Consequently, short wavelength excitation selectively excites the T_n_ segment, which then undergoes ultrafast energy transfer (~1 ps) to afford a TBT-localized excited state.

## 1. Introduction

π-Conjugated oligomers and polymers have delocalized π-electronic systems that make them attractive as active components for applications in molecular-based electronics and opto-electronic devices such as photovoltaic cells, light emitting diodes, and field effect transistors [1,2,3,4,5,6]. Donor–acceptor (D–A) π-conjugated oligomers and polymers are designed to incorporate electron-rich (donor) and electron-deficient (acceptor) repeat units with frontier molecular orbital (MO) energy level alignments suited to promote either excitation energy or charge transfer. The frontier orbital offsets in these molecular systems mimic Type-I and Type-II band offsets in inorganic semiconductor heterojunctions [7,8,9,10,11,12,13]. Solution studies of these organic semiconductor materials enables an understanding of their photophysical, optical, and electronic properties as isolated molecules and provides information on their exited state electronic structure along with effects of a medium dielectric constant and electronic coupling [2]. In these studies, π-conjugated oligomers are used as models, because they provide improved solubility and structural simplicity compared to polymers of large molecular weight and polydispersity [12,14].

There has been interest in research concerning the excited-state electronic structure and dynamics of π-conjugated polymers and oligomers [15,16,17,18,19,20,21,22,23,24,25,26], and some studies in this area focused on comparing the properties of model D–A oligomers and their corresponding co-polymers [2,9]. These studies investigate the effects of excitation energy and conjugation linkage on excited state electronic structures and correlate the results with the device performances of the co-polymers in bulk-heterojunction solar cells [27,28,29]. For instance, a study by Friend and co-workers explored the photophysics of a conjugated system consisting of poly(3-hexylthiophene) (P3HT, 18-repeat units) bonded to a single dioctylfluorene-thiophene-benzothiadiazole-thiophene (F8-TBT) oligomer segment. The results showed that the excited state properties were influenced by the orientation of the linkage between the polymer and the F8-TBT oligomer. A related study compared the properties of an oligomer composed of a cyclopentadithiophene donor and benzothidiazole acceptor (CPDTBT) and its co-polymer (PCPDTBT). The results showed that the nature of the excited state and the polaron pair quantum yield in both the oligomer and polymer were affected by excitation energy, where a higher polaron pair yield was observed with higher energy excitation [28].

We have an ongoing interest in developing donor–acceptor π-conjugated diblock oligomers to investigate their excited state electronic structure and dynamics [8,30,31,32,33,34,35]. These model D–A-type conjugated oligomers provide an opportunity to understand the nature of an initial excited state, wavelength photoselection, exciton evolution, and the nature and rates of interconversion between locally excited (LE) and charge transfer (CT) excited states. These concepts are important in understanding the larger and more complex functional D–A polymers [32,34]. The diblock oligomer design involves linking together fully-conjugated electron-rich and electron-poor segments. In these oligomers, strategies are used to incorporate linkers with different electronic couplings to explore their effect on the above-mentioned properties.

In our recent work, we explored the excited-state electronic structure of conjugated oligomers incorporating either oligothiophene or oligofluorene (donors) linked to a thiophene-benzothiadiazole-thiophene (TBT, acceptor) with varying linkers [32,33,34,35]. The donor and acceptor segments are either strongly coupled (directly linked or linked by ethynyl (-C≡C-)) or weakly coupled (linked by a *trans*-Pt(II) (PBu_3_)_2_ center). In the strongly coupled (directly linked) oligomers with oligothiophene donors that were previously reported (T_4_TBT and T_8_TBT), solvent polarity, electronic coupling-dependent dynamics, and evolution of the excited-state electronic structure were observed [32]. 

The current study is directly related to a recently published article which explored the excited-state properties of two series of diblock oligomers that were adsorbed to the surface of nanostructured TiO_2_ films [35]. The TiO_2_-anchored oligomers were the carboxylic acid (-COOH) forms of the series of ethyl ester functionalized (-COOEt) diblock oligomers shown in Figure 1. 

In the previous study, these oligomers were adsorbed to the TiO_2_ surface via the carboxyl groups, and the photoelectrochemical and excited-state properties of the oligomer-functionalized films were characterized [35]. In the current paper, we report a detailed study of the ester forms of the oligomers (structures in Figure 1) carried out in a solution with solvents of varying polarity. The four π-conjugated oligomers reported herein feature electron-rich oligothiophene (T_n_) segments linked to an electron-poor 4,7-bis(2-thienyl)-2,1,3-benzothiadiazole (TBT) moiety. In these diblock oligomers, the electron donor (T_n_) segment is linked to the acceptor (TBT) by either an ethynyl (-C≡C-) or *trans*-(-C≡C-)_2_Pt(II)(PBu_3_)_2_ acetylide linker. The ester units are present in the structures, because they were the synthetic precursors of the molecules that were anchored onto TiO_2_ in the prior study [35]. The ester groups are essentially “spectators”, as they do not play a significant role in the photophysical results presented herein. The main objectives of this work are to (1) delineate the effect of electronic coupling on the excited-state electronic structure, vis-a-vis locally excited (LE) vs. charge transfer (CT) excited states; (2) explore the effect of excitation wavelength and solvent polarity on the structure and dynamics of the excited states; (3) and examine whether the dynamics of interconversion from the LE and CT excited states are influenced by the linker structure.

## 2. Results 

### 2.1. Structure and Synthesis

The structures of the diblock oligomers are shown in Figure 1. The oligomers contain oligothiophene (T_n_, *n* = 4 or 5) and 4,7-di(thiophen-2-yl)benzo[c][1,2,5]thiadiazole (TBT) segments linked by either an ethynyl (-C≡C-) or a *trans*-(-C≡C-)_2_Pt^II^(PBu_3_)_2_ unit. These two connecting elements are expected to provide a different electronic coupling between the two segments; the ethynyl linker is expected to facilitate electronic communication between the conjugated segments, whereas the *trans*-(-C≡C-)_2_Pt^II^(PBu_3_)_2_ linker decouples the two chromophores [36]. The *trans*-(-C≡C-)_2_Pt^II^(PBu_3_)_2_ linker motif has been used in many previous studies that have explored the role of the organometallic center in the optical, electronic, and optoelectronic properties of π-conjugated oligomers and polymers [37,38]. Carboxyl groups (as esters) are positioned on either side of the oligomers to allow them to be anchored to metal oxide semiconductors with different orientations relative to the T_n_ and TBT moieties. The photophysical and photoelectrochemical properties of the oligomers anchored to metal oxides were explored in a previous study [35]. For synthetic accessibility, ester functionalization of the oligothiophene unit (T_n_) necessitated inclusion of an additional thiophene unit. This is the reason why the oligomers with ester functionality on the T_n_ segment have 5 thiophene repeats, while oligomers with the ester on the TBT moiety contain only 4 thiophene repeats (e.g., ET5TBT vs. T4TBTE). Complete synthesis and characterization details of the diblock oligomers and synthetic intermediates are provided in the Appendix A.

The design concept of this series of diblock oligomers is to explore the interactions between the two conjugated segments in the diblock oligomers as a function of the bridge that links π-conjugated segments. As shown below, to a varying extent, the two oligomer segments (T_n_ and TBT) display distinct absorption features that allow wavelength photoselection to produce an initial (Franck–Condon) excited state that is more or less localized on one of the conjugated segments. A key goal of this work was to delineate if the nature of the initial excited state varies with excitation wavelength and/or the nature of the linker between the segments. 

### 2.2. Photophysical Characterization

Normalized steady-state UV-visible absorption and fluorescence emission spectra of the diblock oligomers in hexane are shown in Figure 1. In the oligomers featuring a T_4_ donor (T4TBTE and T4PtTBTE), the absorption spectra show two well-resolved bands in the 350–550 nm region. The short-wavelength band (λ_max_ = 410 nm in T4TBTE and λ_max_ = 385 nm in T4PtTBTE) is mainly from a π-π* transition in the T_4_ segment and TBT [31], while the long-wavelength band (λ_max_ = 490 nm in T4TBTE and λ_max_ = 530 nm in T4PtTBTE) is due to a π-π* transition with a charge transfer (CT) character arising from the TBT segment (Appendix A). In contrast, in the oligomers featuring an ET_5_ donor (ET5TBT and ET5PtTBT), the bands from the ET_5_ and TBT segments are not as well resolved, because the absorption from the ET_5_ is red-shifted due to increased conjugation due the additional thiophene unit. The short-wavelength, more intense band (λ_max_ = 425 nm in ET5TBT and λ_max_ = 430 nm in ET5PtTBT) is from ET_5_ (Appendix A) while the low energy shoulder band (λ_max_ = 490 nm in ET5TBT and λ_max_ = 525 nm ET5PtTBT) is due to the TBT segment. All the diblock oligomers also show additional features below 400 nm, mainly due to absorption from π-π* transitions in the TBT segment (Appendix A) [34]. The longest wavelength absorption band has a larger oscillator strength in ET5TBT and T4TBTE, which may be due to a charge transfer (CT) interaction between the T_n_ and TBT segments. Similar T_n_ to TBT CT interactions may be weaker in the corresponding Pt-linked oligomers ET5PtTBT and T4PtTBTE due to the intervening metal center that weakens the interaction between the two π-conjugated segments.

The fluorescence of oligomers ET5PtTBT and T4PtTBTE show a broad band centered at 610 nm, which is almost identical to the fluorescence of TBTE in hexane (Appendix A), except red-shifted by ~30 nm. In contrast, the oligomers ET5TBT and T4TBTE show structured emission (λ_max_ = 575 nm), which is like that of the ET5 fluorescence (Appendix A). 

To explore the nature of the emitting states and solvent polarity effects, the UV-visible absorption and fluorescence of the oligomers were studied in solvents with different polarities (Appendix A). In general, all the oligomers show only weak solvatochromism in their absorption spectra; however, the fluorescence spectra broaden and red-shift with increasing solvent polarity. This behavior is characteristic of fluorescence from a CT state and is indicative of an increase in dipole moment in the excited state [30]. Interestingly, in ET5TBT and T4TBTE, there is a distinct change in the emission spectral band shape, from structured in nonpolar hexane to structureless and very red-shifted in polar dichloromethane (DCM). This suggests a significant change in the nature of the excited state with solvent polarity. By contrast, in the Pt-linked oligomers ET5PtTBT and T4PtTBTE (Appendix A), the fluorescence spectra red-shift modestly, and they show a similar trend as the TBTE emission with solvent polarity (Appendix A). This suggests that, for the Pt-linked oligomers, the lowest excited state is localized on the TBT unit [34].

The fluorescence decay of the model and diblock oligomers were studied in selected solvents with an excitation at 410 nm (except for TBTE, 450 nm) and emission at the λ_max_. The lifetimes obtained from analysis of the decay are included in Appendix A, while the decay profiles are provided in Appendix A. In general, all oligomers exhibit bi- or tri-exponential fluorescence decays. The details of the fits are listed in Appendix A; herein, we will just discuss the overall trends as reflected by the average lifetimes <τ>. Thiophene oligomer ET5 has a nearly solvent-independent lifetime (<τ> ~ 300 ps), whereas, for TBTE, <τ> increases with solvent polarity, ranging from 3.4 ns in hexane to 5.4 ns in DCM. For the diblock oligomers, the lifetimes generally become much shorter in a polar solvent (DCM), and they exhibit significant biexponential character. This behavior is similar to that of structurally related organic and (-C≡C-)_2_Pt^II^(PBu_3_)_2_-linked oligomers [32,34,39], and it has been attributed to the interconversion between locally excited and charge transfer excited-state manifolds.

### 2.3. Ultrafast Transient Absorption Spectroscopy

Femtosecond pump-probe transient absorption spectroscopy (fsTA) was used to investigate the dynamics of the excited-state structure and ultrafast energy/charge transfer in the oligomers. To explore wavelength photoselectivity and the effect of solvent polarity, the fsTA studies for the oligomers were performed in hexane (nonpolar) and DCM (polar) using two excitation wavelengths (420 nm and 520 nm) corresponding to the absorption of the T_n_ and TBT segments, respectively. Similar studies were also performed on the model oligomers (ET5 and TBTE). Moreover, all the fsTA data were subjected to a global analysis using the Glotaran (v. 1.5.1) open access software package [40], and the resulting evolution-associated spectra (EAS) and corresponding decay lifetimes, along with the fsTA for all the oligomers, are presented in the Appendix A. In the sections below, the excited-state electronic structure and the effects of solvent polarity and linker type will be discussed using representative examples from the set of TA spectra and corresponding EAS and lifetimes for the entire series. 

#### 2.3.1. Wavelength Photoselection and Effect of Linker Structure

The fsTA spectra and the EAS derived from global analysis for the model oligomers ET5 and TBTE in DCM are presented in (Figure 2) while the results in hexane are included in Appendix A. Generally, the TA and EAS of both oligomers show features of ground state bleach (GSB), stimulated emission (SE), and excited-state absorption (ESA) in agreement with previous reports [32,33,34]. In ET5, the early time TA spectra (0–1 ns) are dominated by an intense, narrow ESA (800 nm) accompanied by GSB (430 nm) and SE (550 nm); these features are characteristic of the T_5_ singlet excited state [41]. At longer delay times (>1 ns), an ESA is seen at ~650 nm that is due to the triplet state. These spectral features are clearly resolved by the two EASs resolved by global analysis (Figure 2c).

The early-time TA of TBTE is dominated by an ESA band at ~725 nm with weaker bands in the visible and near-IR. These are all features of the TBT singlet excited state, which has CT character due to its donor–acceptor–donor structure [32]. The TA features in TBTE persist throughout the experiment time window (7 ns). From the global analysis, the lifetimes of the singlet excited states are ~500 ps (ET5) and ~7 ns (TBTE), which qualitatively agree with the fluorescence lifetime of both oligomers (Appendix A). 

The fsTA and EAS for the diblock oligomers ET5TBT and ET5PtTBT in DCM with excitation wavelengths at 420 nm and 520 nm are shown in Figure 3. Note that 420 nm gives some selectivity for excitation of the oligothiophene segment (T_5_). The early-time fsTA spectra (<1 ps) and the first ESA component (orange solid lines, Figure 3c,g) show distinct features of T_5_-localized excited-state absorption for ET5TBT (850 nm) and ET5PtTBT (970 nm), which are similar to the early-time TA spectra in ET5 (Figure 2a) and those of a previously studied (-C≡C-)_2_Pt^II^(PBu_3_)_2_ complex that was substituted with T_4_ oligomers [34]. 

Interestingly, the evolution of the initially produced T_5_ localized excited state is different in ET5TBT and ET5PtTBT. First, in ET5TBT, there is an ultrafast decay (~1.2 ps) of the SE (580 nm) and the T_5_-based ESA band (850 nm) (first EAS component, Figure 3c). This leads to the appearance of a longer-lived (~170 ps) excited state that is dominated by a broad ESA band in the near-IR at 1050 nm and a tail into the mid-IR (>1600 nm), along with a visibly weaker absorption at 600 nm (blue-dashed line in Figure 3c). These new features are notably different from those of either ET5 or TBTE model oligomers (Figure 2) and are attributed to a delocalized singlet excited state with a CT character. Similar results were previously reported in strongly coupled π-conjugated T_n_TBT oligomers [32]. In those studies, the TA of the oligomers in polar solvents excited at the T_n_ absorption revealed ESA features at 650 nm and a strong peak at 1000 nm (along with a broad band ~1600 nm), which were attributed to CT and delocalized T_n_^+^ (polaron) bands, respectively. However, in the previous study, which featured strongly coupled oligomers, there was no evidence of initial excited-state localization in the thiophene segment at a shorter wavelength excitation as was found here. Similarly, several reports on the TA of oligothiophene films show broad ESA bands at ~1000 nm and ~1500 nm, which were attributed to absorptions of thiophene polarons [31]. 

In contrast, in ET5PtTBT, the T_5_ localized initial excited state evolves (~1.3 ps) to give an ESA with a strong absorption peak at 700 nm and a broad band at 1400 nm; these features are very similar to those seen in TBTE (compare Figure 2b,d), suggesting exciton localization in the TBT segment that occurs following an ultrafast energy transfer from the T_5_ segment. This is consistent with the notion that the (-C≡C-)_2_Pt^II^(PBu_3_)_2_ center decouples the two π-conjugated units. 

At a longer timescale, the features from the delocalized CT state in ET5TBT and the TBT-localized singlet excited state in ET5PtTBT decay in ~200 ps (second ESA component in Figure 3c,g) and give a TA with broad ESA in the 450–800 nm region. The decay time for the second component in both oligomers agrees with their fluorescence emission lifetimes (Appendix A). The longer-time TA feature is attributed to a triplet excited state formed by intersystem crossing (ISC). The triplet excited state features are stronger in ET5PtTBT compared to ET5TBT, consistent with an enhanced ISC in the former due to the presence of a platinum center that induces spin–orbit coupling. 

With excitation at a longer wavelength (520 nm), the TA and EAS spectra for both oligomers qualitatively show similar trends to those obtained at a 420 nm excitation (Figure 3b,d,f,h). At this excitation wavelength, the TBT segment is preferentially excited; however, because of the overlapping absorption of the T_5_ and TBT segments (Figure 1), the early-time TA still shows an absorption peak associated with the T_5_ segment. A closer look, however, shows some differences that suggest that the initially formed excited state has more distinct features of TBT; these differences are more noticeable in the EAS spectra. First, the ground-state bleach (GSB) in both oligomers is narrower and, especially in ET5PtTBT, it touches baseline at 430 nm, which shows a loss in intensity on the blue side of the bleach due to less excitation of the thiophene segment. Second, the SE in ET5PtTBT is significantly quenched, and the 580 nm positive peak appears instantly, which is the case in the model TBTE, while, in ET5TBT, this peak gets broader and stronger. Third, in both diblocks, the ESA ~700–900 nm shows a stronger absorption on the blue side, and, in the case of ET5PtTBT, the sharp 970 nm peak is much weaker. All these differences show that there is some wavelength photoselectivity in the excitation of the diblock segments. Interestingly, regardless of the excitation wavelength or which segment is preferentially excited, the initial excited state in each diblock oligomer evolves in the same manner to the one observed at 420 excitations, which is exciton localization on TBT in ET5PtTBT and a charge transfer to populate a delocalized CT state in ET5TBT. At last, at a delay time of >7 ns, the singlet excited state undergoes intersystem crossing to populate the triplet excited state like the excitation at 420 nm.

#### 2.3.2. Effect of Solvent Polarity on Excited-State Structure and Dynamics

Given that the oligomers studied here exhibit charge-transfer properties in their excited states, we also systematically explored the effect of solvent polarity on the excited state spectroscopy and dynamics. To simplify the discussion, in the manuscript we present data for T4TBTE in two solvents (non-polar hexane and polar DCM). All the oligomers in Figure 1 were examined in these two solvents, and the entire set of data is in the Appendix A.

The fsTA difference spectra and evolution-associated spectra from global analysis (EAS) for T4TBTE in hexane and DCM with an excitation at 420 nm are compared in Figure 4 (corresponding spectra obtained with 520 nm excitation are in Appendix A). In hexane (Figure 4a,c), excitation produces an excited state that persists for ~1 ns. This excited state features GSB at 500 nm, SE at 600 nm, and a broad ESA band with λ_max_ ~ 1000 nm. The SE shows a vibronic structure similar to the steady-state fluorescence emission in hexane (Figure 1, top panel), and its rapid (~5 ps) red-shift is likely due to conformational relaxation [41]. Global analysis of the data (EAS, Figure 4c) reveals two similar spectral components due to the singlet excited state and then a weaker spectral component with a long lifetime (25 ns) that is the triplet excited state. The lifetime of the singlet state from global analysis (862 ps) is very similar to the average fluorescence lifetime (930 ps, Appendix A). The takeaway message is that diblock oligomer T4TBTE in a non-polar solvent exhibits a single singlet excited state that is produced promptly following excitation. The spectral features are distinct from those of either of the individual segments (T_4_ or TBT), suggesting that they are in a delocalized state. The singlet relaxes within a few ns and produces the triplet state in a good yield as assessed by the amplitude of the triplet absorption features.

In a more polar DCM solvent, the fsTA spectral dynamics are distinctly different from those in hexane. The most prominent difference is that there are two distinct excited states that are present at early times following excitation. These are most easily discerned by reference to the evolution-associated spectra (EAS) obtained from global analysis (Figure 4d). Here it is seen that, at very early times following excitation, the spectrum difference is very similar to that observed in hexane. However, this species is replaced very rapidly (~1.8 ps) by a new excited state that has a GSB at 500 nm, a new ESA peak at 620 nm, and a second broad near-IR ESA band (λ_max_ ~ 1000 nm) that extends into the mid-IR (λ > 1400 nm). This longer-lived state decays with a lifetime of ~250 ps, which is in reasonable agreement with the average fluorescence lifetime in DCM (180 ps). There is a very weak ESA at the longest decay times that may be due to a triplet state; however, the low amplitude of the signal suggests that, in the more polar solvent, the triplet yield is low.

## 3. Discussion

This paper describes a comprehensive study of a series of four different diblock oligomers that contain oligothiophene (T_n_) and TBT segments linked with either acetylene (-C≡C-) or platinum acetylide (-C≡C-)_2_Pt^II^(PBu_3_)_2_ units. We have compared/contrasted the photophysical properties of the oligomers, including fs-transient absorption spectra, in different solvents and under different excitation wavelengths. The objective was to determine the effect of the linker between the T_n_ and TBT segments and solvent on the structure and dynamics of the excited states. We also sought to identify whether wavelength photoselection could be used to selectively excite the T_n_ or TBT segments. Key results that support the main conclusions are included and discussed in the main text, while the comprehensive set of data for the entire series of oligomers in different solvents and different excitation wavelengths is included in the Appendix A.

Each of the oligomers features two primary, low energy absorption bands which can be assigned to transitions mainly localized on the T_n_ or TBT segments. The absorption bands are best resolved for the platinum-linked oligomers (ET5PtTBT and T4PtTBTE). Moreover, the two absorption features become less well resolved for the oligomers that feature the T_5_ segment (ET5TBT and ET5PtTBT). All the oligomers feature fluorescence bands that are moderately to strongly Stokes-shifted from the absorption bands. The fluorescence of the (-C≡C-)-linked oligomers (ET5TBT and T4TBTE) exhibits the strongest solvatochromism, suggesting that their excited states have a substantial degree of charge-transfer character. This notion is also supported by the solvent-dependent transient absorption studies, which show that, in polar solvents, the (-C≡C-)-linked oligomers display strong and broad near-IR absorption features that are associated with the T_n_^+^ (polaron) absorption.

Excitation wavelength dependence studies give evidence that photoselection produces very short-lived excited states that are mainly localized on the oligomer segment that is initially excited. This behavior is most distinct for ET5PtTBT (Figure 3e), where a 420 nm excitation leads to the appearance of a narrow peak at 950 nm due to the T_5_ segment at very early times but decays rapidly (τ ~ 1.3 ps). This feature is less prominent with an excitation at 520 nm, consistent with the photoselection of the lower energy TBT segment at longer wavelengths. For ET5TBT, the evidence for wavelength photoselection is less clear. This difference may be due to the fact that, for this oligomer, the initially formed excited state is delocalized, and, thus, the excitation wavelength has less of an effect.

The key observations of this work can be summarized by the excited-state schemes that are shown in Figure 5a,b. The most complex behavior is observed for the (-C≡C-)-linked diblock oligomers, ET5TBT and T4TBTE, shown in the diagram in Figure 5a. In a non-polar solution, the excitation of these oligomers produces a singlet excited state, which decays mainly by fluorescence and non-radiative pathways, ultimately affording a moderate yield of a triplet excited state. The transient absorption of this state is distinct from that of either model oligomer (ET5 or TBT), suggesting that the excited state is delocalized across both segments of the oligomer. We assign this state to an LE designation (locally excited state). By contrast, in a more polar solvent (e.g., DCM), there is evidence for the formation of an initially excited state that rapidly evolves (~1 ps) to a second excited state that persists for much longer. We hypothesize that, in the more polar solvent, the excited state takes on a charge-transfer (CT) character, with the T_n_ segment acting as the donor and the TBT segment as the acceptor. Although TA studies were carried out in only two solvents, it is evident from the fluorescence spectra of ET5TBT and T4TBTE in different solvents (Appendix A) that the energy of the CT state decreases with increasing solvent polarity. The important point is that due to the relatively strong coupling between the T_n_ and TBT segments that is afforded by the (-C≡C-) linker, there is not strong evidence for excited state localization on either segment, even at very early times following excitation. The nature of the excited states evolves depending on the solvent, varying from a delocalized LE state in non-polar medium to a CT state in more polar solvents.

By contrast, due to weak coupling between the T_n_ and TBT segments in Pt-linked oligomers ET5PtTBT and T4PtTBTE, we observe behavior that suggests that short wavelength excitation produces a T_n_ localized excited state that undergoes ultrafast energy transfer (~1.3 ps) to afford a TBT-localized excited state that persists for longer times. This behavior is most evident by an inspection of the TA spectra of ET5PtTBT with an excitation at 420 nm (Figure 3e,g). At an early time, there is a prominent excited state absorption peak at 970 nm, which decays in several ps, leaving a species that has a TA spectrum that is very similar to that of the TBT model oligomer (TBTE, Figure 2b,d). Excitation at a longer wavelength (520 nm, Figure 3f,h) directly produces the TBT-localized excited state. Regardless of the excitation wavelength, there is a long-lived triplet state that is produced following decay of the singlet states. The excited state behavior of the Pt-linked diblock oligomers is summarized in Figure 5b. Here, short wavelength excitation populates a singlet excited state that is localized on the T_n_ segment. This state decays via an ultrafast intramolecular energy transfer process to afford a TBT-localized singlet excited state. At much longer timescales, the TBT singlet state decays via radiative and non-radiative pathways, including intersystem crossing to a TBT-localized triplet with moderate yield. This latter pathway is facilitated by spin–orbit coupling enhanced by the heavy metal Pt center [42]. 

## 4. Experimental Methods

### 4.1. Synthesis and Characterization

Synthesis details are provided in the Appendix A. ^1^H and ^13^C NMR spectra were recorded on a Bruker Ascend TM spectrometer (operates at 500 MHz for ^1^H and 125 MHz for ^13^C) in CDCl_3_. Chemical shifts (δ) are reported in parts per million (ppm) using references to CHCl_3_. High-resolution mass spectrometry was performed with a Bruker Daltonics Ultraflextreme MALDI-TOF/TOF mass spectrometer in the Chemistry Department at the University of Texas, San Antonio, TX, USA.

### 4.2. Photophysical Analysis

UV–visible absorption spectra were acquired on a Shimadzu UV-2600 dual beam absorption spectrophotometer. Steady-state photoluminescence and time-correlated single-photon counting (TCSPC) instrumentation and methods used in this work are described in a previous article [33]. For emission lifetime measurements on all oligomers, an excitation wavelength of 410 nm was used, except for TBTE, where 450 nm was used for excitation.

### 4.3. Electrochemistry

Cyclic voltammetry measurements were carried out using a CH Instruments electrochemical workstation in DCM solvent with 0.1 M tetrabutyl ammonium hexafluorophosphate electrolyte. Glassy carbon was used as a working electrode, Pt wire as an auxiliary electrode, and Ag/Ag^+^ in acetonitrile as a quasi-reference electrode, and the scan rate was 100 mV·s^−1^. Ferrocene (Fc/Fc^+^) was used as internal standard, *E_½_* (Fc/Fc^+^) = 0.17 V vs. Ag/Ag^+^ and HOMO and LUMO values were computed from the electrochemical potentials by assuming that E(Fc/Fc^+^) = −4.8 V on the vacuum scale [43,44].

### 4.4. Femtosecond Transient Absorption Spectroscopy

Femtosecond pump-probe transient absorption spectroscopy (fs-TA) instrumentation is described in detail in a previous article [34,45]. The pump beam from the optical parametric amplifier was tuned to generate 420 nm, 490 nm, and 520 nm pulses for excitation. Transient absorption data were analyzed using Surface Xplorer PRO program (v 4) from Ultrafast Systems (Sarasota, FL, USA) and global analysis was performed using the open access software Glotaran (v 1.5.1) [40].

## 5. Conclusions

This paper reports the full synthetic details and photophysical properties of a novel series of diblock oligomers that feature oligothiophene (T_n_) and thiophene-benzothiadiazole-thiophene (TBT) segments coupled either by (-C≡C-) or (-C≡C-)_2_Pt^II^(PBu_3_)_2_ linkers. The photophysics of the oligomers is complex, exhibiting dependence on the excitation wavelength (photoselection) as well as on solvent polarity. One key finding is that the properties of the diblocks depend strongly on the linker. For ET5TBT and T4TBTE, it is clear that the (-C≡C-) linker provides relatively strong electronic coupling between the two segments. The oligomers display relatively muted photoselection behavior; however, the excited state properties vary strongly with solvent polarity. Taken together, the results reveal that, in all solvent media, the lowest excited state is a delocalized state which involved both conjugated segments. In non-polar solutions, the excited state has minimal charge transfer character, but, in more polar solvents, the excited state has CT character, with the T_n_ segment acting as donor and the TBT segment as the acceptor.

By contrast, in the Pt-linked oligomers ET5PtTBT and T4PtTBTE, there is relatively weak coupling between the T_n_ and TBT segments. Short wavelength excitation gives photoselection, producing an excited state that is largely localized on the T_n_ segment. This state evolves via ultrafast energy transfer (~1 ps) to afford the lowest excited state that is mainly localized on the TBT segment. The Pt-atom enhanced the spin–orbit coupling, giving rise to more efficient population of a long-lived TBT-localized triplet excited state.

## Data Availability

Data are contained within the article and Appendix A.

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
