# Peer review of "Characterization of Excited-State Electronic Structure in Diblock π-Conjugated Oligomers with Adjustable Linker Electronic Coupling"

_molecules, 2024, doi:10.3390/molecules29112678_

Round 1
Reviewer 1 Report
Comments and Suggestions for Authors
The English language is good, although the manuscript contains few mistakes (some of them are mentioned in the report).
Reviewer 2 Report
Comments and Suggestions for Authors
This paper describes the synthetic routes to a series of oligomers and gives a full structural characterisation of them. The optical properties of the materials are further elucidated by tests including UV, fluorescence, etc. The excited state properties vary strongly with solvent polarity. I think the manuscript can be accepted with minor modifications.
Please see the document for some specific modifications and suggestions.
